# HIV replication and tuberculosis risk among people living with HIV in Europe: A multicohort analysis, 1983–2015

Andrew Atkinson[1,2,3º], David Kraus[1,2,4º], Nicolas Banholzer[1], Jose M. Miro[5,6], Peter Reiss[7,8], Ole Kirk[9,10], Cristina Mussini[11], Philippe Morlat[12,13], Daria Podlekareva[9,14], Alison D. Grant[15,16], Caroline Sabin[17], Marc van der Valk[7,8,18], Vincent Le Moing[19], Laurence Meyer[20], Remonie Seng[20], Antonella Castagna[21], Niels Obel[10], Anastasia Antoniadou[22], Dominique Salmon[23], Marcel Zwahlen[1], Matthias Egger[1,24,25], Stephane de Wit[26], Hansjakob Furrer[2‡], Lukas Fenner[1‡*], **The Opportunistic Infections Project Working Group of the Collaboration of Observational HIV Epidemiological Research Europe (COHERE) in EuroCoord[¶]**

1 Institute of Social and Preventive Medicine, University of Bern, Bern, Switzerland, 2 Department of Infectious Diseases, Bern University Hospital, University of Bern, Bern, Switzerland, 3 Infectious Diseases Division, Washington University School of Medicine in St Louis, St Louis, MO, United States of America, 4 Department of Mathematics and Statistics, Masaryk University, Brno, Czech Republic, 5 Infectious Diseases Service, Hospital Clinic–IDIBAPS, University of Barcelona, Barcelona, Spain, 6 CIBERINFEC, Instituto de Salud Carlos III, Madrid, Spain, 7 Stichting HIV Monitoring, Amsterdam, The Netherlands, 8 Amsterdam UMC, location University of Amsterdam, Global Health, and Amsterdam Institute for Global Health and Development, Amsterdam, The Netherlands, 9 Centre of Excellence for Health, Immunity and Infections (CHIP), Copenhagen University Hospital, Rigshospitalet, Copenhagen, Denmark, 10 Department of Infectious Diseases, Rigshospitalet, Copenhagen University Hospital, Copenhagen, Denmark, 11 Clinic of Infectious Diseases, University of Modena and Reggio Emilia, Modena, Italy, 12 ISPED, INSERM Bordeaux Population Health Research Center, Team PHARes, UMR 1219, University of Bordeaux, Bordeaux, France, 13 Service de Médecine Interne et Maladies Infectieuses, Centre Hospitalier Universitaire de Bordeaux (CHU), Bordeaux, France, 14 Department of Respiratory and Infectious Diseases, Copenhagen University Hospital, Bispebjerg, Denmark, 15 Department of Clinical Research, London School of Hygiene & Tropical Medicine, London, United Kingdom, 16 Africa Health Research Institute, School of Laboratory Medicine & Medical Sciences, College of Health Sciences, University of KwaZulu-Natal, KwaZulu-Natal, South Africa, 17 Institute for Global Health, University College London, London, United Kingdom, 18 Department of Infectious Diseases, Location University of Amsterdam, Amsterdam Institute for Immunology & Infectious Diseases, Infectious Diseases Program, Amsterdam UMC, Amsterdam, The Netherlands, 19 CHU de Montpellier, Université de Montpellier, Montpellier, France, 20 Department of Public Health and Epidemiology, AP-HP, Bicêtre Hospital, INSERM CESP U1018, Paris-Saclay University, Le Kremlin Bicêtre, France, 21 Department of Infectious Diseases, IRCCS San Raffaele Scientific Institute, Milan, Italy; Vita-Salute San Raffaele University, Milan, Italy, 22 University General Hospital Attikon, School of Medicine, National and Kapodistrian University of Athens, Athens, Greece, 23 Department of Infectious Diseases, Institut Fournier, Paris, France, 24 Population Health Sciences, Bristol Medical School, University of Bristol, Bristol, United Kingdom, 25 Centre for Infectious Disease Epidemiology and Research, Faculty of Health Sciences, University of Cape Town, Cape Town, South Africa, 26 Department of Infectious Diseases, Saint Pierre University Hospital, Université Libre de Bruxelles, Brussels, Belgium

º These authors contributed equally to this work.
‡ HF and LF also contributed equally to this work.
¶ The complete membership of the collaboration can be found in the Acknowledgments.
* lukas.fenner@unibe.ch



**Data Availability Statement:** Data used for the analysis will generally not be publicly available for ethical and legal reasons but can be made available based on the approval by the chair of the executive

committee of COHERE (contact: christine.
schwimmer@u-bordeaux.fr).

**Funding:** The COHERE study group has received
unrestricted funding from: Agence Nationale de
Recherches sur le SIDA et les Hépatites Virales
(ANRS), France; HIV Monitoring Foundation, The
Netherlands; and the Augustinus Foundation,
Denmark. COHERE received funding from the
European Union Seventh Framework Programme
[grant no. FP7/2007–2013] under EuroCoord grant
agreement no. 260694. Research reported in this
publication was supported by the Swiss National
Science Foundation [grant no. 324730_149792].
AA and DK were supported by the Swiss National
Science Foundation [grant no. 324730_149792].
NB, ME, and LF are supported by the National
Institute of Allergy and Infectious Diseases (NIAID)
through grant no. 5U01-AI069924-05. ME is
supported by special project funding from the
Swiss National Science Foundation [grant no.
32FP30-189498]. JMM received a personal 80:20
research grant from Institut d'Investigacions
Biomèdiques August Pi i Sunyer (IDIBAPS),
Barcelona, Spain, during 2017–24. All other
authors report no competing interests. The funders
had no role in study design, data collection, data
analysis, data interpretation, or writing of the
report.

**Competing interests:** The authors have declared
that no competing interests exist.

## Abstract

### Introduction

HIV replication leads to a change in lymphocyte phenotypes that impairs immune protection against opportunistic infections. We examined current HIV replication as an independent risk factor for tuberculosis (TB).

### Methods

We included people living with HIV from 25 European cohorts 1983–2015. Individuals <16 years or with previous TB were excluded. Person-time was calculated from enrolment (baseline) to the date of TB diagnosis or last follow-up information. We used adjusted Poisson regression and general additive regression models.

### Results

We included 272,548 people with a median follow-up of 5.9 years (interquartile range [IQR] 2.3–10.9). At baseline, the median CD4 cell count was 355 cells/µL (IQR 193–540) and the median HIV-RNA level 22,000 copies/mL (IQR 1,300–103,000). During 1,923,441 person-years of follow-up, 5,956 (2.2%) people developed TB. Overall, TB incidence was 3.1 per 1,000 person-years (95% confidence interval [CI] 3.02–3.18) and was four times higher in patients with HIV-RNA levels of 10,000 compared with levels <400 copies/mL in any CD4 stratum. CD4 and HIV-RNA time-updated analyses showed that the association between HIV-RNA and TB incidence was independent of CD4. The TB incidence rate ratio for people born in TB-endemic countries compared with those born in Europe was 1.8 (95% CI 1.5–2.2).

### Conclusions

Our results indicate that ongoing HIV replication (suboptimal HIV control) is an important risk factor for TB, independent of CD4 count. Those at highest risk of TB are people from TB-endemic countries. Close monitoring and TB preventive therapy for people with suboptimal HIV control is important.

## Introduction

Tuberculosis (TB) is the leading cause of death from an infectious disease, along with HIV/AIDS [1]. HIV infection and its resulting immunodeficiency is the most important risk factor for TB, and in many lower-income countries TB is the most common opportunistic infection (OI) [2, 3]. Antiretroviral therapy (ART) has improved the prognosis of HIV infection and markedly reduced the risk of TB [4]. In several observational studies, successful ART has been associated with a lower incidence of TB in low-income countries with high TB burden as well as in affluent countries with low levels of TB transmission [5, 6]. In Western European countries with a low TB incidence, most TB patients are foreign-born, reflecting higher rates of TB infection in their country of birth and subsequent development of active TB in the country of migration [7, 8].

While the risk for TB and other OIs in HIV infection is commonly related to immunodeficiency as measured by circulating CD4 positive T-cells in peripheral blood (CD4 cell count),

people living with HIV at a given CD4 count may have a higher risk for OIs if there is HIV replication as measured by plasma HIV-RNA [9–12]. There is increasing evidence that immune activation due to ongoing HIV replication is an additional important factor of immune disturbance leading to important changes in lymphocyte phenotypes and quality of immune response [13]. In two randomized trials, the early initiation of ART among individuals with high CD4 counts was beneficial regarding the incidence of OIs, including TB [14, 15]. While certain OIs occur almost exclusively in people with very low CD4 counts [16], active TB is observed in every stage of HIV-infection. Even in those with high CD4 counts TB incidence is much higher than in the population of people without HIV [17]. In South Africa, HIV-RNA levels have been shown to be a risk factor for TB independent of CD4 count [11].

The past 15 years have seen a massive scale-up of ART and a decline in AIDS-related deaths worldwide [18], especially with the change in recommendations in 2015 for all people with HIV to receive ART [14, 19]. Despite these positive changes in the public health epidemiology of HIV/TB, understanding the interplay between HIV and TB remains critical for identifying high-risk groups in need of TB preventive therapy (TPT) and for developing optimal clinical and public health strategies. Using the large Collaboration of Observational HIV Epidemiological Research Europe (COHERE) database from 1984–2015, which includes a substantial number of people with both HIV and TB (>5,000), we aimed to delineate the role of current HIV replication as an independent risk factor for TB.

## Methods

### Study setting and study design

COHERE in EuroCoord was a collaboration of over 40 observational cohorts covering 32 European countries within the EuroCoord network of excellence [20]. Each cohort submitted data in a standardized format (the HIV Collaboration Data Exchange Protocol, see http://www.hicdep.org) to coordinating centres at the Copenhagen HIV Programme, Denmark, or the Institut de Santé Publique d'Epidémiologie et de Développement (Bordeaux School of Public Health), Bordeaux, France. The Regional Coordinating Centres ensured adherence to strict data quality assurance guidelines and performed data checks. Submitted data included information on individuals' characteristics (age, gender, geographical origin, and mode of HIV acquisition), use of ART (type of drugs, and dates of start and stop of each drug), dates and results of all CD4 cell counts and plasma HIV-RNA viral load assessments, dates and types of all AIDS-defining events, and information on mortality. Data on testing for latent TB and on TPT were not available. We included people aged 16 years or older with HIV who were enrolled in care between the first quarter of 1983 and the second quarter of 2015 and had no history of previous TB. The selection of patients from 29 COHERE cohorts with TB data is shown in a flow diagram (S1 Fig); four cohorts with too few eligible patients (<350) or lack of specific TB drug/endpoint information were excluded. The data was first accessed for research purposes in June 2019.

### Statistical analysis

The first TB occurrence was defined to be the time of the person's first diagnosis of pulmonary or extrapulmonary tuberculosis. Person-time was defined from enrolment in the respective cohort to first TB occurrence, death, or date of last follow-up visit. A person on ART was defined as a patient taking combined ART at any time during follow-up. Incident TB was defined as any TB diagnosis two months or more after ART start. The diagnosis of TB was made by the treating clinicians according to local diagnostic criteria.

We used Poisson regression models to assess TB incidence per 1,000 person-years and corresponding rate ratios with an offset for the period, stratified by the levels of baseline CD4 cells/µL (0–100, 101–350, >350) and HIV-RNA copies/mL (0–399, 400–10,000, >10,000) and geographical region of birth (Europe, sub-Saharan Africa, Asia, Latin America, North Africa and the Middle East). Models were additionally adjusted for age, gender, geographic region of birth, and mode of HIV acquisition (sex between men and women, sex between men [MSM], injection drug use [IDU], other). We extended regression models including time-updated CD4 cell counts and HIV-RNA levels and using splines to investigate effects in detail. For illustrative purposes, we used the fitted models to predict the incidence for a reference category of men who were aged 35 years, who had acquired HIV following heterosexual sex, and who were on ART.

All analyses were performed in R version 3.2.4 [21].

### Ethics statement

Ethical approval was applied for and granted for the research from the appropriate body in the host country (local ethics committees or institutional review boards) of the cohort contributing the data to COHERE [20]. All cohorts participating in COHERE adhere to local ethical, data management and confidentiality standards.

### Results

Follow-up data for 272,548 people from 25 European cohorts (S1 Table) were included in the analysis, resulting in a total follow-up time of 1,923,441 person-years. The median follow-up time was 5.9 years (interquartile range [IQR] 2.3–10.9). Baseline characteristics (at enrolment into cohort in Europe) of the participants are shown in Table 1: 27% of participants were female, 65% were born in Europe (including Eastern Europe), and 46% were on ART. The median CD4 cell counts was 355 cells/µL at baseline, and 13% had a HIV-RNA levels below 400 copies/mL. There were 5,956 first episodes of TB over follow-up, resulting in an overall TB incidence of 3.1 per 1,000 person-years (95% confidence interval [CI] 3.0–3.2). TB occurred at a median of 1.5 years (IQR 0.2–4.5 years) after enrolment in the respective cohort.

In the Poisson regression model stratifying by CD4 count and HIV-RNA levels (Table 1), TB incidence was significantly associated with the following baseline characteristics: gender, HIV acquisition mode other than sex between men, participant geographic origin (particularly in those born in Africa, Asia, and Latin America), CD4 cell count <350 vs.≥350, HIV-RNA ≥400 vs. <400 copies /ml, and not being on ART at baseline (Table 1). In addition, the incidence rate ratio for people born in Asia and sub-Saharan Africa compared to those from Europe were 2.5 (95% CI 2.0–3.0) and 2.4 (95% CI 2.2–2.6), respectively. More generally, the incidence rate ratio of those of non-European compared to European origin was 1.8 (95% CI 1.5–2.2). There was no significant interaction between geographic origin and HIV-RNA or CD4 count (each p = 0.34).

Time-updated analyses of the CD4 cell count and HIV-RNA levels, fixed at different CD4 levels for a reference participant category, showed that the association between HIV-RNA and TB incidence was independent of CD4 cell counts (Fig 1). The trajectory of the HIV-RNA curves was similar at all CD4 cell levels, with slopes becoming steeper at levels above 10,000 HIV-RNA copies/mL and reaching four times higher incidence levels than those with HIV-RNA levels below 400 copies/mL. A similar association was observed in people from all geographic regions (Fig 2). In all CD4 count and viral load categories, the incidence of TB was highest among people of Asian and sub-Saharan African origin (Fig 2): 81/1,000 and 82/1,000 person-years, respectively, for the highest risk category. Time-updated CD4 cell counts (cells/µL)

**Table 1. Individual characteristics (at enrolment into cohort) of 272,548 people with HIV from 25 European cohorts, with and without tuberculosis (TB) after enrolment.**

| Characteristic n (%) | Total in study | People with TB | People without TB | IRR (95% CI) | p-value |
|---|---|---|---|---|---|
| **Number of people** | 272,548 | 5,956 (2.2) | 266,592 (97.8) | | |
| **Time period** | | | | | <0.001 |
| 1984–1995 | 24,165 (8.9) | 1,001 (4.1) | 23,164 (95.9) | | |
| 1996–2015 | 248,383 (91.1) | 4,955 (2.0) | 24,3428 (98.0) | | |
| **Female gender** | 73,370 (26.9) | 2,021 (33.9) | 71,349 (26.8) | 0.80 (0.75–0.86) | <0.001 |
| **Age**, years, median (IQR) | 36.0 (30.1–43.2) | 34.9 (29.8–41.2) | 36.0 (30.1–43.2) | 0.99 (0.99–0.99) | <0.001 |
| **Mode of HIV acquisition** | | | | | |
| Sex between men and women | 99,893 (36.7) | 2,914 (48.9) | 96,979 (36.4) | 1.0 | |
| Sex between men | 113,715 (41.7) | 1,174 (19.7) | 112,541 (42.2) | 0.41 (0.37–0.44) | <0.001 |
| Injection drug use | 31,823 (11.7) | 1,245 (20.9) | 30,578 (11.5) | 1.03 (0.95–1.12) | 0.46 |
| Other | 9,035 (3.3) | 251 (4.2) | 8,784 (3.3) | 0.77 (0.67–0.89) | <0.001 |
| *Missing, n (%)* | *18,082 (6.6)* | *372 (6.2)* | *17,710 (6.6)* | - | |
| **CD4**, cells/µL, median (IQR) | 355 (193–540) | 243.0 (100–414) | 358 (196–542) | | <0.001 |
| <100 | 33,530 (12.3%) | 1,254 (21.1%) | 32,376 (12.1%) | 1.0 | |
| 100–350 | 86,109 (31.6%) | 2,102 (35.3%) | 83,914 (31.5%) | 0.31 (0.29–0.32) | <0.001 |
| >350 | 122,472 (44.9%) | 1,653 (27.8%) | 120,819 (45.3%) | 0.11 (0.10–0.12) | <0.001 |
| *Missing, n (%)* | *30,530 (11.2)* | *947 (15.9)* | *29,583 (11.1)* | - | |
| **HIV viral loads**, $\log_{10}$ copies/mL, median (IQR) | 4.3 (3.1, 5.0) | 4.8 (4.0, 5.3) | 4.3 (3.1, 5.0) | | <0.001 |
| **HIV viral loads**, copies/mL | | | | | |
| <400 | 36,266 (13.3) | 304 (5.1) | 35,962 (13.5) | 1.0 | |
| 400–10,000 | 45,532 (16.7) | 594 (10.0) | 44,938 (16.9) | 1.69 (1.54–1.84) | <0.001 |
| >10,000 | 120,052 (44.0) | 2,641 (44.3) | 117,411 (44.0) | 3.8 (3.51–4.08) | <0.001 |
| *Missing, n (%)* | *70,738 (25.9)* | *2,417 (40.6)* | *68,281 (25.6)* | - | |
| **Region of origin** | | | | | |
| Europe | 176,088 (64.6) | 3,263 (54.8) | 172,825 (64.8) | 1.0 | |
| Africa | 26,946 (9.9) | 1,240 (20.8) | 25,706 (9.6) | 2.43 (2.24–2.64) | <0.001 |
| Asia | 3,464 (1.3) | 129 (2.2) | 3,335 (1.3) | 2.45 (2.00–2.99) | <0.001 |
| Latin America | 9,681 (3.6) | 248 (4.2) | 9,433 (3.5) | 1.74 (1.50–2.02) | <0.001 |
| N Africa/Middle East | 4,119 (1.5) | 129 (2.2) | 3,990 (1.5) | 1.40 (1.13–1.72) | 0.002 |
| Other or unknown | 52,250 (19.2) | 947 (15.9) | 51,303 (19.2) | 1.35 (1.24–1.47) | <0.001 |
| **On ART** | 123,322 (45.2) | 2,027 (34) | 121,295 (45.5) | 0.76 (0.72–0.81) | <0.001 |
| **Follow-up time**, years, median (IQR) | 5.9 (2.3–10.9) | 1.5 (0.2–4.5) | 6.1 (2.4–11.0) | - | |

ART, antiretroviral therapy; CI, confidence interval; IRR, incidence rate ratio

stratified by geographic origin showed similar curves in all regions, but people originating from Europe had lower TB incidence at all CD4 cell count levels compared to those from all other regions (S2 Fig).

## Discussion

We analyzed data from almost 300,000 people with HIV enrolled into care in 25 large HIV cohorts in Europe from 1983 to 2015. Overall TB incidence was at 3.10 per 1,000 person-years. We found that individuals with increasing HIV replication, as determined by plasma HIV-RNA levels, had an increasing risk of TB, independent of CD4 cell count and all other variables. The risk of TB was particularly high in people born in Africa and Asia, reflecting the higher prevalence of TB in these regions.

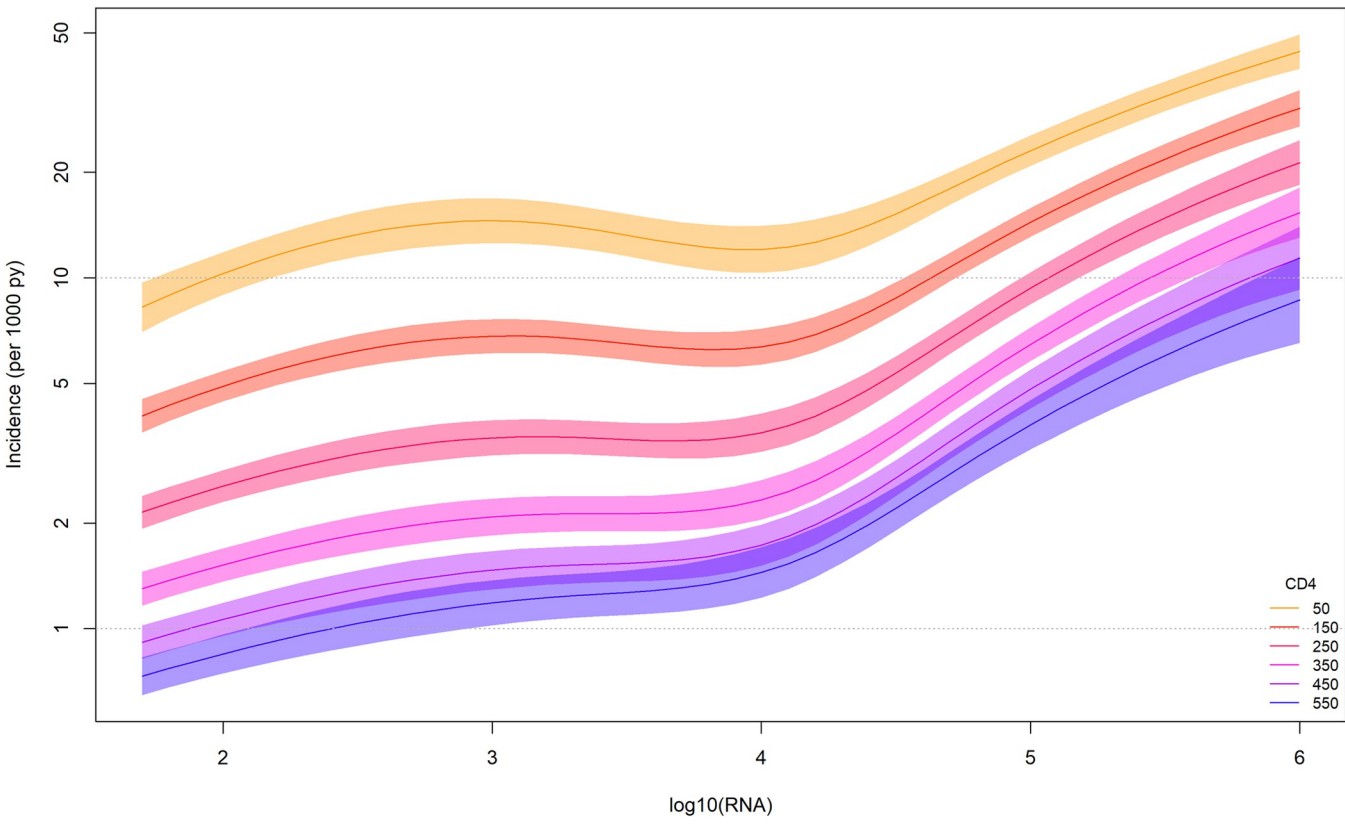

**Fig 1. Tuberculosis (TB) incidence rate per 1,000 person-years for a reference population estimated at any given HIV-RNA levels, based on time-updated measurements and displayed at different CD4 levels.** Patient reference category: heterosexual, male, 35 years old, on ART, born in Europe. Shaded areas corresponding to 95% confidence intervals.

The large collaborative HIV cohort dataset in Europe allowed us to show that HIV viral load provides useful information for predicting higher risk of TB after ART initiation, independent of CD4 cell count. This finding might be explained by the fact that HIV replication itself is associated with a disturbed immune system and impaired protection against progression from infection to disease [13]. Underlying immunological processes include increased cell turnover, cell activation, and cytokine release [11, 13]. This is also evidenced by a decreased immune response to vaccines such as yellow fever and influenza in patients with ongoing HIV replication with high HIV-RNA levels [22, 23]. We have shown that detectable HIV-RNA, which indicates ongoing HIV replication, is a critical laboratory parameter for identifying populations at higher risk for TB, along with CD4 count and place of birth. All current guidelines [14, 19] recommend that all people with HIV receive ART, regardless of their CD4 cell count at the time of diagnosis. However, this was not the case during the period analyzed in this study. In Europe, widespread access to effective combination ART began in 1996 and has dramatically reduced the number of deaths among people living with HIV [20, 24].

In this large European dataset with a diverse migrant population, we observed that TB risk depends on a person's geographic origin. In Europe, the overall incidence of TB has been decreasing since 2000, but notification rates vary across the region, with the highest rates reported in Eastern Europe [25]. The same decline has been found across European HIV cohorts and in the Swiss HIV Cohort Study [26, 27]. In contrast, some European countries have reported an increase in TB incidence among people with HIV because of increasing

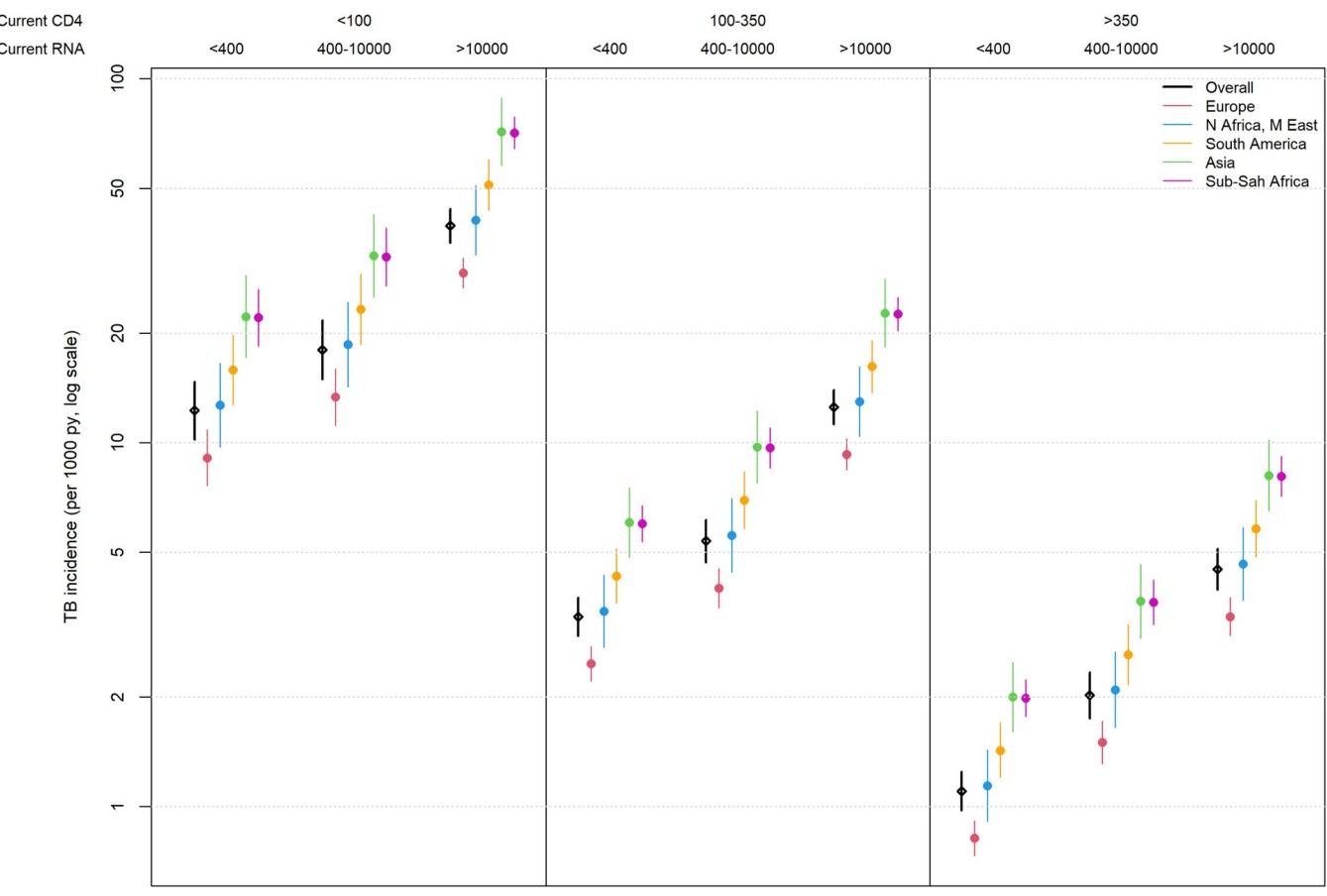

**Fig 2. TB incidence stratified by CD4 cell count (cells/µL), HIV-RNA viral load (copies/mL), and geographic origin for a reference population.** TB incidence rates per 1,000 person-years estimated from fitted Poisson regression models. Patient reference category: 35 years of age, male, heterosexual, on ART.

numbers of migrants from lower-income countries with high TB incidence and possibly because of delayed access to care [28–30]. The current war in Ukraine, with internal displacement of citizens and migration of refugees, may further increase the burden of TB in Europe, particularly drug-resistant TB [31]. Indeed, migrants to Europe with HIV/TB co-infection may not receive optimal care and follow-up in low-incidence countries [32]. Differences also exist between Western and Eastern Europe in the delivery of TB and HIV care, including integration of TB and HIV care and access to MDR-TB drugs [33]. In general, risk factors for TB in people with HIV include not being on ART, the origin of the person (e.g., sub-Saharan Africa and Eastern Europe), but also low CD4 cell counts and higher HIV viral loads [30]. However, the latter study did not examine the possibility of there being an interaction between the CD4 cell count and HIV viral load for the risk of TB.

People with HIV who have detectable HIV viral loads at any time are at increased risk of active TB by reactivation of latent TB, reinfection, or rapid progression after recent infection, regardless of their CD4 cell counts. Therefore, in addition to early initiation of ART or optimization of ART, these individuals may particularly benefit from the administration of TPT to reduce the risk of TB in persons with latent TB, especially in migrant populations originating from TB-endemic countries and possibly higher exposure to TB. Prevention of active TB disease by TPT is a critical component of the WHO End TB Strategy [34] and has been shown to be highly effective in the Western European setting [12]. WHO recommendations for settings

with a high burden of TB and HIV state that adults, adolescents, children, and infants with HIV should receive TPT, irrespective of the degree of immunosuppression and even if latent TB infection testing is unavailable [35]. The positive impact of ART on reducing the incidence of TB in people with HIV has been well established [6]. In any case, people at high risk identified in our study should be considered for latent TB screening and TPT if active TB has been excluded and should be closely monitored during HIV care to reduce the risk of treatment interruptions and eventually ensure early diagnosis of TB.

A limitation of our study is that the data date back many years, spanning 1983–2015. However, the current epidemiology of HIV/TB in Europe no longer allows for detailed and stratified analyses, as only a few new TB cases have been diagnosed in recent years [27, 28] and most of these are in people with a late HIV diagnosis [36]. While this changing epidemiology is welcome, it does mean that only these older data allow detailed analyses of risk factors for TB based on CD4 cell counts, HIV RNA viral loads, and person's birth origin. Such analyses provide valuable information for current management and monitoring in high-risk populations, particularly in the context of ongoing wars and immigration. Other limitations include the limited number of people with HIV followed in Eastern Europe, which has a different TB epidemiology [37], and that there may be residual confounding (homelessness, instable social conditions, alcohol use) that was not captured in our dataset. An important strength of the study is that the data come from a well-established, large-scale, collaborative HIV cohort project that has collected high-quality data with little missing information [20].

In conclusion, understanding the interplay between HIV and TB is paramount to developing clinical management and effective public health strategies to reduce the burden of TB in people with HIV. Our results show that higher levels of ongoing HIV replication are associated with a higher risk of TB, regardless of CD4 cell count. Individuals with latent TB, often originating from countries at high risk of TB infection, particularly require effective ART, close clinical monitoring to ensure undetectable HIV-RNA levels, and a low threshold for administration of TPT once active TB has been excluded.

## Supporting information

**S1 Table. Breakdown of participants in the study by cohort.**
(PDF)

**S1 Fig. Selection of the study population.**
(PDF)

**S2 Fig. TB incidence rate per 1,000 person-years for a reference population, based on time-updated measurements and stratified by CD4 counts (reference: 35 years of age, male, heterosexual, on ART).** A: All regions are shown unless the region was unknown. B: Only sub-Saharan Africa, Asia, and Europe are shown for clarity.
(PDF)

## Acknowledgments

We thank all individuals whose data were used in this study. We also would like to thank all who contributed to recording and entering data and preparing and sending it to the COHERE collaboration.

**Steering Committee—Contributing Cohorts:** Ali Judd (AALPHI), Robert Zangerle (AHIVCOS),Giota Touloumi (AMACS), Josiane Warszawski (ANRS CO1 EPF/ANRS CO11 OBSERVATOIRE EPF), Laurence Meyer (ANRS CO2 SEROCO), François Dabis (ANRS CO3 AQUITAINE), Murielle Mary Krause (ANRS CO4 FHDH), Jade Ghosn (ANRS CO6

PRIMO), Catherine Leport (ANRS CO8 COPILOTE), Linda Wittkop (ANRS CO13 HEPA-VIH), Peter Reiss (ATHENA), Ferdinand Wit (ATHENA), Maria Prins (CASCADE), Heiner Bucher (CASCADE), Diana Gibb (CHIPS), Gerd Fätkenheuer (Cologne-Bonn), Julia Del Amo (CoRIS), Niels Obel (Danish HIV Cohort), Claire Thorne (ECS, NSHPC), Amanda Mocroft (EuroSIDA), Ole Kirk (EuroSIDA), Christoph Stephan (Frankfurt), Santiago Pérez-Hoyos (GEMES-Haemo), Osamah Hamouda (German ClinSurv), Barbara Bartmeyer (German ClinSurv), Nikoloz Chkhartishvili (Georgian National HIV/AIDS), Antoni Noguera-Julian (CORISPE-cat), Andrea Antinori (ICC), Antonella d'Arminio Monforte (ICONA), Norbert Brockmeyer (KOMPNET), Luis Prieto (Madrid PMTCT Cohort), Pablo Rojo Conejo (CORISPES-Madrid), Antoni Soriano-Arandes (NENEXP), Manuel Battegay (SHCS), Roger Kouyos (SHCS), Cristina Mussini (Modena Cohort), Jordi Casabona (PISCIS), Jose M. Miró (PISCIS), Antonella Castagna (San Raffaele), Deborah_Konopnick (St. Pierre Cohort), Tessa Goetghebuer (St Pierre Paediatric Cohort), Anders Sönnerborg (Swedish InfCare), Carlo Torti (The Italian Master Cohort), Caroline Sabin (UK CHIC), Ramon Teira (VACH), Myriam Garrido (VACH). David Haerry (European AIDS Treatment Group)

**Executive Committee:** Stéphane de Wit (Chair, St. Pierre University Hospital), Jose Mª Miró (PISCIS), Dominique Costagliola (FHDH), Antonella d'Arminio-Monforte (ICONA), Antonella Castagna (San Raffaele), Julia del Amo (CoRIS), Amanda Mocroft (EuroSida), Dorthe Raben (Head, Copenhagen Regional Coordinating Centre), Geneviève Chêne (Head, Bordeaux Regional Coordinating Centre). Paediatric Cohort Representatives: Ali Judd, Pablo Rojo Conejo.

**Regional Coordinating Centres**: Bordeaux RCC: Diana Barger, Christine Schwimmer, Monique Termote, Linda Wittkop; Copenhagen RCC: Casper M. Frederiksen, Dorthe Raben, Rikke Salbøl Brandt.

**Project Leads and Statisticians**: Juan Berenguer, Julia Bohlius, Vincent Bouteloup, Heiner Bucher, Alessandro Cozzi-Lepri, François Dabis, Antonella d'Arminio Monforte, Mary-Anne Davies, Julia del Amo, Maria Dorrucci, David Dunn, Matthias Egger, Hansjakob Furrer, Marguerite Guiguet, Sophie Grabar, Ali Judd, Ole Kirk, Olivier Lambotte, Valériane Leroy, Sara Lodi, Sophie Matheron, Laurence Meyer, Jose Mª Miró, Amanda Mocroft, Susana Monge, Fumiyo Nakagawa, Roger Paredes, Andrew Phillips, Massimo Puoti, Eliane Rohner, Michael Schomaker, Colette Smit, Jonathan Sterne, Rodolphe Thiebaut, Claire Thorne, Carlo Torti, Marc van der Valk, Linda Wittkop.

## Author Contributions

**Conceptualization:** Andrew Atkinson, David Kraus, Marcel Zwahlen, Matthias Egger, Hansjakob Furrer, Lukas Fenner.

**Data curation:** Andrew Atkinson, David Kraus, Jose M. Miro, Peter Reiss, Ole Kirk, Cristina Mussini, Philippe Morlat, Daria Podlekareva, Alison D. Grant, Caroline Sabin, Marc van der Valk, Vincent Le Moing, Laurence Meyer, Remonie Seng, Antonella Castagna, Niels Obel, Anastasia Antoniadou, Dominique Salmon, Stephane de Wit, Hansjakob Furrer.

**Formal analysis:** Andrew Atkinson, David Kraus, Nicolas Banholzer.

**Funding acquisition:** Marcel Zwahlen, Hansjakob Furrer, Lukas Fenner.

**Investigation:** Andrew Atkinson, Lukas Fenner.

**Methodology:** Andrew Atkinson, David Kraus, Marcel Zwahlen, Lukas Fenner.

**Resources:** Jose M. Miro, Peter Reiss, Ole Kirk, Cristina Mussini, Philippe Morlat, Daria Podlekareva, Alison D. Grant, Caroline Sabin, Marc van der Valk, Vincent Le Moing,

Laurence Meyer, Remonie Seng, Antonella Castagna, Niels Obel, Anastasia Antoniadou, Dominique Salmon, Matthias Egger, Stephane de Wit, Hansjakob Furrer.

**Supervision:** Hansjakob Furrer, Lukas Fenner.

**Visualization:** Andrew Atkinson, David Kraus, Nicolas Banholzer.

**Writing – original draft:** Andrew Atkinson, David Kraus, Nicolas Banholzer, Hansjakob Furrer, Lukas Fenner.

**Writing – review & editing:** Andrew Atkinson, David Kraus, Nicolas Banholzer, Jose M. Miro, Peter Reiss, Ole Kirk, Cristina Mussini, Philippe Morlat, Daria Podlekareva, Alison D. Grant, Caroline Sabin, Marc van der Valk, Vincent Le Moing, Laurence Meyer, Remonie Seng, Antonella Castagna, Niels Obel, Anastasia Antoniadou, Dominique Salmon, Marcel Zwahlen, Matthias Egger, Stephane de Wit, Hansjakob Furrer.

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
