## [Decision Letter · Decision Letter 0]

2 Jul 2024

PONE-D-24-10899HIV replication and tuberculosis risk among people living with HIV in Europe: a multicohort analysis, 1983-2015PLOS ONE

Dear Dr. Fenner,

Thank you for submitting your manuscript to PLOS ONE. After careful consideration, we feel that it has merit but does not fully meet PLOS ONE’s publication criteria as it currently stands. Therefore, we invite you to submit a revised version of the manuscript that addresses the points raised during the review process.

We look forward to receiving your revised manuscript.

Kind regards,

Maemu Petronella Gededzha, Ph.D

Academic Editor

PLOS ONE

Journal Requirements:

- https://doi.org/10.7448/IAS.20.1.21327

In your revision ensure you cite all your sources (including your own works), and quote or rephrase any duplicated text outside the methods section. Further consideration is dependent on these concerns being addressed.

Reviewers' comments:

Reviewer's Responses to Questions

**Comments to the Author**

1. Is the manuscript technically sound, and do the data support the conclusions?

Reviewer #1: Yes

Reviewer #2: Yes

2. Has the statistical analysis been performed appropriately and rigorously? 

Reviewer #1: Yes

Reviewer #2: Yes

3. Have the authors made all data underlying the findings in their manuscript fully available?

Reviewer #1: No

Reviewer #2: Yes

4. Is the manuscript presented in an intelligible fashion and written in standard English?

Reviewer #1: Yes

Reviewer #2: Yes

5. Review Comments to the Author

Reviewer #1: The manuscript provides an excellent analysis of HIV cohort spanning over many decades from European countries. Below are some of the comments to authors:

1. There are several typos that require attention i.e line 106, line 124

2. Data availability is provided under two websites. However the COHERE website is inaccessible or does not work.

3. Ethics statement is provided but does not include how data was protected and confidentiality maintained. A follow-up statement would be helpful.

4. It is not clear how TB diagnosis was done and/or a local diagnostic criteria could be provided.

5. History of BCG vaccination is important and information on its availability or implications for the study is key. Some countries use it as part of their TB prevention strategy.

6. A general comment on the availability of ART would be useful since the cohort started in 1983. Which year was ART made available to the participants?

7. Other risk factors identified could be reported i.e alcohol, nutrition, smoking, non-adherence etc

8. Future implications of TPT based on HIV viral load could be added to the recommendations. Some previous recommendations were based on CD4 count.

Reviewer #2: TB is still the leading cause of death among PLHIV even in countries that have achieved optimal ART roll out. The authors show the impact of suboptimal HIV control on TB incidence and that this relationship between detectable viral load and TB incidence is maintained at all CD4 strata.

These findings are important for countries with large TB and HIV burden and further emphasize the need for frequent viral load monitoring among PLHIV started on ART and the importance of additional TB preventive measures e.g., TB preventive therapy among persons living with HIV.

6. PLOS authors have the option to publish the peer review history of their article (what does this mean?). If published, this will include your full peer review and any attached files.

Reviewer #1: **Yes: **Lesibana Malinga

Reviewer #2: No

---

## [Author Response · Author response to Decision Letter 0]

29 Aug 2024

The response letter has been uploaded as a separate document.

---

## [Decision Letter · Decision Letter 1]

30 Sep 2024

HIV replication and tuberculosis risk among people living with HIV in Europe: a multicohort analysis, 1983-2015

PONE-D-24-10899R1

Dear Dr. FENNER,

We’re pleased to inform you that your manuscript has been judged scientifically suitable for publication and will be formally accepted for publication once it meets all outstanding technical requirements.

Kind regards,

Maemu Petronella Gededzha, Ph.D

Academic Editor

PLOS ONE

Additional Editor Comments (optional):

Reviewers' comments:

Reviewer's Responses to Questions

**Comments to the Author**

1. If the authors have adequately addressed your comments raised in a previous round of review and you feel that this manuscript is now acceptable for publication, you may indicate that here to bypass the “Comments to the Author” section, enter your conflict of interest statement in the “Confidential to Editor” section, and submit your "Accept" recommendation.

Reviewer #1: All comments have been addressed

2. Is the manuscript technically sound, and do the data support the conclusions?

Reviewer #1: Yes

3. Has the statistical analysis been performed appropriately and rigorously? 

Reviewer #1: Yes

4. Have the authors made all data underlying the findings in their manuscript fully available?

Reviewer #1: Yes

5. Is the manuscript presented in an intelligible fashion and written in standard English?

Reviewer #1: Yes

6. Review Comments to the Author

Reviewer #1: (No Response)

7. PLOS authors have the option to publish the peer review history of their article (what does this mean?). If published, this will include your full peer review and any attached files.

Reviewer #1: **Yes: **Lesibana Anthony Malinga

---

## [Editor Report · Acceptance letter]

15 Oct 2024

PONE-D-24-10899R1 

PLOS ONE

Dear Dr. Fenner, 

I'm pleased to inform you that your manuscript has been deemed suitable for publication in PLOS ONE. Congratulations! Your manuscript is now being handed over to our production team.

Kind regards, 

on behalf of

Dr. Maemu Petronella Gededzha 

Academic Editor

PLOS ONE